# Characterization of *Microbotryum lychnidis-dioicae* Secreted Effector Proteins, Their Potential Host Targets, and Localization in a Heterologous Host Plant

**DOI:** 10.3390/jof10040262

**Published:** 2024-03-30

**Authors:** Ming-Chang Tsai, Michelle T. Barati, Venkata S. Kuppireddy, William C. Beckerson, Grace Long, Michael H. Perlin

**Affiliations:** 1Department of Biology, College of Arts and Sciences, University of Louisville, Louisville, KY 40292, USA; mingchangtsai@hotmail.com (M.-C.T.); mail2swathi.k@gmail.com (V.S.K.); william.beckerson@louisville.edu; 2Department of Medicine, Division of Nephrology & Hypertension, School of Medicine, University of Louisville, Louisville, KY 40202, USA; michelle.barati@louisville.edu

**Keywords:** *Microbotryum lychnidis-dioicae*, fungal effector protein, protein-protein interaction, CASP-like protein 2C1, COP9 signalosome subunit 5a/5b, trichome

## Abstract

*Microbotryum lychnidis-dioicae* is an obligate fungal species colonizing the plant host, *Silene latifolia*. The fungus synthesizes and secretes effector proteins into the plant host during infection to manipulate the host for completion of the fungal lifecycle. The goal of this study was to continue functional characterization of such *M. lychnidis-dioicae* effectors. Here, we identified three putative effectors and their putative host-plant target proteins. MVLG_02245 is highly upregulated in *M. lychnidis-dioicae* during infection; yeast two-hybrid analysis suggests it targets a tubulin α-1 chain protein ortholog in the host, *Silene latifolia*. A potential plant protein interacting with MVLG_06175 was identified as CASP-like protein 2C1 (CASPL2C1), which facilitates the polymerization of the Casparian strip at the endodermal cells. Proteins interacting with MVLG_05122 were identified as CSN5a or 5b, involved in protein turnover. Fluorescently labelled MVLG_06175 and MVLG_05122 were expressed in the heterologous plant, *Arabidopsis thaliana*. MVLG_06175 formed clustered granules at the tips of trichomes on leaves and in root caps, while MVLG_05122 formed a band structure at the base of leaf trichomes. Plants expressing MVLG_05122 alone were more resistant to infection with *Fusarium oxysporum*. These results indicate that the fungus might affect the formation of the Casparian strip in the roots and the development of trichomes during infection as well as alter plant innate immunity.

## 1. Introduction

The smut fungus *Microbotryum lychnidis-dioicae* is an obligate plant parasite primarily infecting *Caryophyllaceae* species and resulting in smutted anthers [1]. In the current study the plant host is *Silene latifolia*. Similar to other smut-fungal species, *M. lychnidis-dioicae* has a diphasic life cycle with haploid and dikaryotic stages. The infection starts as the diploid teliospore germinates and undergoes meiosis to yield haploid cells after landing on the host flower. The fimbriae of a haploid cell extend to search for a mate. When two haploid cells of opposite mating type make contact with each other, they fuse to become dikaryotic hyphae via conjugation and start penetrating host tissues with the use of appressoria. The diploid hyphae may reach the roots, stay dormant during the winter, and migrate to the flower primordia in the spring. When the hyphae reach the stamen, karyogamy occurs and the dikaryotic hyphae become diploid teliospores. The hyphae also utilize the plant cellular machinery to turn the anther into a structure filled with the teliospores, and the infection cycle restarts [2]. Although female flowers do not have anthers, the fungus can inhibit the development of gynoecium while enhancing that of anther filament from a rudimentary stamen so that the teliospore sac still replaces the structures of anthers on female flowers [3].

Phytopathogens infecting and colonizing plants include prokaryotic bacteria, eukaryotic fungi, oomycetes, and nematodes. They have evolved dynamic genomes encoding protein products transported or extruded out of cells to facilitate the degradation of plant cell wall, absorption of nutrients, and/or modulation of the host. The corresponding genes are associated with synthesis of secondary metabolites such as mycotoxins and the effector proteins. Effector proteins, also referred to as small-secreted proteins (SSPs), have appeared as the primary molecular interactors with plant hosts to modify plant structure, metabolism, and defensive responses in order to benefit the lifecycle of phytopathogens in plant-microbe interactions [4]. The effector proteins often evolve at rapid rates, leading to high specificity for their plant hosts. There are also low degrees of conserved amino-acid sequences among fungal effectors [5]. Comparing effector proteins with non-effector proteins, the effector proteins carry higher percentages of cysteine residues but lower percentages of serine and tryptophan residues in the amino acid sequence based on the predicted models of EffectorP. Furthermore, a broader distribution of net protein charges is also a distinct feature of the fungal effector proteins [6]. The fungal effector proteins can be approximately separated into the apoplastic (extracellular) and cytoplasmic effector proteins based on their destinations in the plant host. The apoplastic effector proteins are secreted into the apoplast or xylem, while the cytoplasmic effector proteins enter the cytoplasm of the cells. The former tend to contain multiple cysteine residues. However, this feature is not universal. The disulfide bridges between cysteine residues could reinforce the stability of effector proteins in the apoplast, a generally oxidative environment, also full of host proteases [7].

Bacterial phytopathogens evolved the type III secretion system to directly deliver effector proteins into the tissues of plant hosts. With respect to eukaryotic fungi and oomycetes, certain species have also evolved special structures to deliver effector proteins into the hosts. When the hyphae penetrate the plant cell wall, some biotrophic fungal species and oomycetes secrete effector proteins into plant cells through the haustoria which develop at the terminal region of hyphae. The haustorium is a specialized feeding structure surrounded by the plasma membrane of the plant cell [8,9]. Between the plant cell membrane and the pathogen cell wall there is the extrahaustorial matrix. Pathogenic effector proteins are released into the matrix, and they may be translocated into the cytoplasm of the host cells by endocytosis while binding to receptors on the plant cell membrane. However, plant cells may also release proteases into the matrix to hydrolyze pathogenic effector proteins [9]. Once the pathogenic effector proteins enter the cytoplasm of the host cells, they may start modifications of host metabolism and structures as well as trigger the plant immune responses.

Our team has previously shown that effector proteins of *M. lychnidis-dioicae* could potentially interact with synaptotagmins of the host plant, membrane proteins associated with the vesicle trafficking and signal transduction, and with the cellulose synthase interactive protein 1 (CSI1), a regulator of microtubule and anther development [10]. In the current study we further identified the plant protein interactors of three fungal effector proteins MVLG_02245, MVLG_05122, and MVLG_06175, the locations of potential protein-protein interactions in host plant tissues, and any significant phenotype changes in plant hosts caused by the latter two effector proteins.

## 2. Materials and Methods

### 2.1. cDNA Library Construction

Total RNAs were isolated from a variety of stages of the *M. lychnidis-dioicae* life cycle, including in vitro stages and different stages of infection of male and female *S. latifolia*. The samples included RNA sequences of both the fungus and the plant host. The samples from each life stage were pooled and sent to CD Genomics (Shirley, NY, USA) for reverse transcription and establishment of the cDNA library within a prey vector, pGADT7 (Clonetech, pGADT7 AD Vector Information, protocol No. PT3249-5, version No. 010312), for use in yeast two-hybrid screening (Y2H).

### 2.2. Cloning the Genes MVLG_02245, MVLG_06175, and MVLG_05122

The gene sequences for predicted fungal effectors MVLG_02245, MVLG_06175, and MVLG_05122, are available in the JGI Fungal Genome database, as part of the genome of *M. lychnidis-dioicae* [11] (URL accessed on 19 February 2019 https://mycocosm.jgi.doe.gov/Micld1/Micld1.home.html). Additional bioinformatic comparisons were carried out using the MVLG_02245 sequence of *M. lychnidis-dioicae*, retrieved from JGI for BLASTX 2.10.0+ analyses against the PacBio genomes QPIF00000000 *M. silenes-dioicae* 1303 a2 from *Silene dioica* and GCA_900015495.1 *M. violaceum var. paradoxa* from *Silene paradoxa* 1252 a1, URL accessed on 4 January 2019. Blastn2.10.0+ and Blastp2.10.0+ were performed using the NCBI local-alignment tool, SequenceServer2.0.0.rc8, query submitted 4 January 2019. Pfam 32 and HMMER 3.1b1 suite (https://hmmer.org, URL accessed on 5 April 2019) were used to screen the JGI translated gene model MVLG_02245 for any known protein families using a cutoff value of 1 × 10^−3^ for significance. The coding DNA and translated protein sequences were also used in blast analysis against the NCBI online database to identify any similar proteins. SignalP4.1 was used to predict the secretion of coding sequences obtained from JGI, as well as to determine the signal peptide and functional protein regions of the corresponding translated protein sequence. PONDR and IUPred2A (URLs https://www.pondr.com and https://iupred2a.elte.hu accessed 30 January 2017) were used to screen for ordered protein folding of the protein sequence. Expression data were obtained from [12,13] to verify production of the MVLG_02245 transcript.

The upstream primers for each effector gene included a sequence recognized by the restriction enzyme EcoRI at its 5′ end, whereas the downstream primer carried one recognized by BamHI. Primers were designed to exclude the signal sequence during PCR amplification. The conditions of the PCR cycle were 94 °C of initial denaturation temperature for 4 min, followed by 35 cycles of 94 °C for 30 s, 60 °C for 30 s, and 72 °C for 45 s. The final extension was 72 °C for 5 min. Gel electrophoresis in 1.5% agarose (Agarose Unlimited USB Corp., Cleveland, OH, USA) was used to separate the amplified gene sequences. After PCR, each fungal effector gene was cloned by TOPO TA Cloning into the vector pCR^®^ 2.1 (Thermo Fisher Scientific, Inc., Waltham, MA, USA) and transformed into *Escherichia coli* strain TOP-10 cells, according to the recommendations of the manufacturer. The addition of 250 μL of SOC broth and 1 h shaking followed by incubation at 37 °C allowed expression of its introduced DNA, after which 120 μL of aliquot was spread to Luria broth (LB) agar plates (0.5% yeast extract, 1% sodium chloride, 0.5% tryptone, and 2% agar). The media contained 200 μg/mL ampicillin and X-β-gal, as the colorimetric indicator of β-galactosidase activity; this allowed screening for plasmids with inserts, for which the colonies would be white. The transcription of β-galactosidase would be interrupted if the *MVLG_02245*, *MVLG_06175* or *MVLG_05122* were successfully inserted into the multiple cloning sites of the vector. The vector pCR^®^ 2.1 containing the cloned fungal effector genes were propagated in *E. coli* and later extracted. by an alkaline lysis protocol. Purified plasmids were then digested with restriction enzymes EcoRI and BamHI and the sequences of the inserted genes were determined by Sanger sequencing (Eurofins, Louisville, KY, USA). The constructs were propagated in *E. coli* strain DH5α (Bethesda research Laboratories, Bethesda, MD, USA).

### 2.3. Yeast Transformation

Frozen-EZ Yeast Transformation II™ (Zymo Research Corp., Irvine, CA, USA) was applied to transform gene constructs into competent yeast cells. The bait vector pGBKT7 carried the three fungal effector genes, while the prey vector pGADT7 cDNA library carried the cDNA library prepared by CD Genomics (Shirley, NY, USA). These genetically modified bait and prey vectors were transformed into yeast strain AH109 and Y187, respectively, with certain modifications in the procedure. The Kit includes solutions EZ 1, 2 and 3. To make competent AH109 and Y187 strains, yeast cells were grown shaking in 10 mL YPD broth (1% yeast extract, 2% peptone, 10% dextrose, and 0.1% kanamycin) at 30 °C overnight. The ideal absorbance at 660 nm (OD660) to harvest the competent cells was between 0.8 and 1.0 (between 5 × 10^6^ and 2 × 10^7^ cells/mL). The yeast culture was centrifuged at 1000 rpm for 4 min. The pellet was resuspended in 10 mL of solution EZ 1 after discarding the supernatant, followed by repeating the centrifugation of the yeast culture and resuspending the pellet with solution EZ 2. In order to enhance transformation efficiency, the cell pellet was resuspended in 0.5 mL of solution EZ 2 rather than 1 mL, thereby increasing cell concentration prior to use in transformation. Otherwise, the protocol followed the recommendations of the manufacturer. Competent yeast cells incubated with desired DNAs were spread onto plates containing selection medium (leucine drop-out medium for pGADT7 and tryptophan drop-out medium for pGBKT7), and the plates were incubated at 30 °C for 2–4 days. Yeast colonies with the desired constructs were harvested and maintained at −80 °C for long-term storage.

### 2.4. Yeast Secretion Trap Assay

Signal peptides lead the newly synthesized protein products to reside on the cell membrane and certain organelles such as Golgi apparatus. They also guide some proteins to be secreted from the cell to the surrounding medium or environment. As a result, it is necessary to verify that fungal protein effectors, *MVLG*_02245, MVLG_06175 and MVLG_05122 were secreted from fungal cells, and a yeast secretion trap (YSP) test [14] was used. *Saccharomyces cerevisiae* strain SEY 6210 in the test cannot synthesize invertase catalyzing the hydrolysis of sucrose to fructose and glucose. To rescue this defect in sucrose utilization, vector pYSTO-0 was employed as a test of predicted secretion signal peptides in putatively secreted proteins. The vector contains the coding region of *SUC2*, but the gene lacks the start codon and a portion encoding the signal sequence. *SUC2* will not express invertase in the absence of the start codon and, unless a functional secretion signal is provided, invertase will remain in the cytosol. Therefore, SEY 6210 cells do not have the ability to utilize sucrose as a sole carbon source. Additionally, the vector encodes the gene *LEU2* for production of leucine, allowing selection of successfully transformed colonies in a medium lacking leucine.

The secretion signal sequences of *MVLG_02245*, *MVLG_06175* and *MVLG_05122* were amplified by PCR and inserted upstream of the *SUC2* gene in the pYSTO-0 vector by restriction enzyme digestion and ligation. Each genetically constructed vector was transformed into *E. coli* strain DH5α (Bethesda research Laboratories, Bethesda, MD, USA), and later was extracted from bacterial transformants and then transferred into the SEY 6210 yeast strain.

### 2.5. Yeast Two-Hybrid Screening

Yeast-two hybrid screening (Y2H) with Gal4 transcription factor can be used to identify the interactions between biotrophic fungal proteins and host plant proteins [15,16,17]. The protocol of Matchmaker™ Pretransformed Library User Manual (Clontech Laboratories, Inc., Mountain View, CA, USA) was applied for Y2H. Genes of interest and genes of potential target are cloned into “bait” and “prey” vectors, respectively. The bait vector was pGBKT7 (Clonetech, pGBKT7 Vector Information, protocol No. PT3248-5, version No. PR8Y2643), encoding a Gal4 DNA binding domain (BD) and a gene required for tryptophan biosynthesis, while the prey vector was pGADT7 (Clonetech, pGADT7 AD Vector Information, protocol No. PT3249-5, version No. 010312), encoding a Gal4 DNA activation domain (AD) and a gene for leucine biosynthesis. The fungal effector genes were cloned into the bait vector, and a cDNA library acquired from flower tissues of *S. latifolia* infected by *M. lychnidis-dioicae* was cloned into the prey vector. For directed yeast two-hybrid experiments with specific genes, the bait and prey vector were digested with restriction enzymes EcoRI and BamHI (with the appropriate buffer) to become linear. The purified fragments of *MVLG_02245*, *MVLG_06175*, and *MVLG_05122* collected from the restriction digestion were connected to the linearized bait vector pGBKT7 by T4 DNA ligase. These vectors were propagated in *E. coli* strain DH5α (Bethesda research Laboratories, Bethesda, MD, USA).

Subsequently, the bait vector was transformed into the yeast strain AH109 and the prey vector was transformed into the strain Y187. For selection of the desired yeasts in the medium, yeast strain AH109, which bears the bait vector pGBKT7, is able to synthesize tryptophan and proliferate in tryptophan-deficient medium; in the case of Y187 which bears the prey vector pGADT7, the cells are able to synthesize leucine and proliferate in a leucine-deficient medium. Furthermore, strain AH109 contains chromosomally encoded reporter genes which include *HIS3* gene required for histidine synthesis, *ADE2* required for adenine synthesis, and *MEL1* expressing α-galactosidase. α-galactosidase cleaves a chromogenic substrate 5-bromo-4-chloro-3-indolyl alpha-d-galactopyranoside (X-α-gal) in the medium. The cleavage of X-α-gal releases a blue biomarker resulting in blue yeast colonies. After genes of interest are fused with BD in the bait vector and genes of potential target are fused with AD in the prey vector, if in the offspring yeasts the two protein products interact physically with each other and thus BD and AD are in close proximity, the reporter genes will initiate expression and the protein-protein interactions yield proliferation of yeasts on a nutrient-deficient medium. All drop-out media were made with sterile deionized water, glucose at 20 g/L, yeast nitrogen base at 6.7 g/L, drop out mix 2 g/L, and agar at 15 g/L. Additionally, the 3-amino-1,2,4-triazole (3-AT) was added to the medium, which inhibited the potential leaky expression of the histidine marker.

A positive and several negative controls were included in the Y2H spot test. The positive control is the protein-protein interaction between pGBKT7-p53 (expressing p53 protein) and pGADT7-T(expressing T antigen), marked as BD-p53+AD-T. The negatives controls are (1) vectors alone/without bait and prey protein genes (i.e., BD and AD, respectively), (2) vectors without bait and prey protein genes, and two yeast strains mated (BD+AD), (3) p53 in BD alone and T in AD alone. (i.e., BD-p53 and AD-T, respectively), and (4) one of the two mating yeast strains carries bait or prey vectors with no inserts, including BD-5122+AD, BD+AD-CSN5a/5b, BD-CSN5a/5b+AD, and BD+AD-5122.

### 2.6. Gibson Assembly to Construct MVLG_05122 Tagged with a Cyan Fluorescent Protein Gene and MVLG_06175 Tagged with a mCherry Protein Gene

Unlike the gene construction by restriction digestion and ligation for Y2H, Gibson assembly [18] was used to create plasmids bearing the desired constructs encoding versions of MVLG_06175 and MVLG_05122 tagged with fluorescent proteins. The vector pRI-101AN used here encodes a kanamycin resistance gene, the cauliflower mosaic virus 35S promoter (CaMV 35S), and a NOS transcriptional terminator. CaMV 35S is a robust promoter to enhance the expression of the inserted genes of interest in plants. Vector pRI-101AN was cut with restriction enzymes BamHI and NdeI and underwent ethanol precipitation.

The fungal effector gene *MVLG_05122* without its signal sequence and a cyan fluorescent protein (CFP) gene were incorporated into the vector pRI-101AN. Additionally, nine nucleotides encoding glycine-glycine-serine served as a linker and were added in-frame between *MVLG_05122* and the CFP gene for a certain level of flexibility between the respective proteins in the resulting fusion protein. To yield additional expression constructs, the fungal effector *MVLG_06175* with and without its signal sequence and the mCherry protein gene were also incorporated into the pRI-101AN vector. NEBuilder^®^ HiFi DNA Assembly Master Mix (New England BioLabs, Inc., Ipswitch, MA, USA) was used to accomplish Gibson assembly. Selection was on LB agar containing kanamycin (50 μg/mL). The resulting bacterial colonies were screened for plasmid content, underwent DNA extraction, and sequencing of plasmids was used to confirm the presence of the respective fungal gene and the in-frame gene for its tag.

### 2.7. Agrobacterium Transformation through Electroporation

Competent cells of bacterium *Agrobacterium tumefaciens* were prepared based on the Pikaard Lab protocol (https://pikaard.lab.indiana.edu/protocols/protocols/agrobacterium-growth-and-transformation.html; URL accessed on 20 April 2016). The transformation was conducted in a BioRad micropulser electroporator with voltage of 2.5 kV using the 25 mF capacitor and at 400-ohm settings [19]. DNA extraction from *Agrobacterium tumefaciens* strain EHA105 colonies and PCR were conducted to verify the presence of the *MVLG_05122ΔSP-CFP*, *MVLG_06175ΔSP-mCherry*, and *MVLG_06175-mCherry* constructs in putative transformants. *Agrobacterium* clones confirmed to carry the constructed vector would be used to infect *Arabidopsis* and deliver the respective constructs.

### 2.8. Arabidopsis thaliana and Growth Conditions

*Arabidopsis thaliana* ecotype Col-0 was used as the wild-type background in the current study (kindly provided by Dr. Mark Running, University of Louisville). The plant seeds underwent surface-sterilization and were spread onto and cultivated on 1/2× MS solid media (Murashige & Skoog, Phytotechnology Laboratories, Cat No: M524) which contained 0.05% MES buffer {2-(N-morpholino) ethanesulfonic acid, ThermoFisher, Pittsburgh, Pennsylvania, USA} and 0.8% agar as well as kanamycin (50 μg/mL) for selection of transgenic plants. The media needed to be adjusted to pH 5.7. The seeds were kept at 4 °C for 2 to 3 days and were transferred to 20–24 °C for 10 days for further germination. Seedlings were transplanted to pots with soil (Sungro Horticulture propagation mix, Premium Horticultural Supply, Louisville, KY, USA, cat no. 5232601). The environment for the growth of *Arabidopsis* was at 22 °C, with 68% relative humidity (RH), light intensity 120 μmol m^−2^s^−1^ and with a 16 h/8 h day/night cycle.

### 2.9. Floral Dipping Transformation of Arabidopsis thaliana Mediated by Agrobacterium

The transformation was based upon the protocol of Zhang et al. [20]. After confirming, via by PCR and DNA sequencing, that the *A. tumefaciens* strains carried genes of interest, a single *Agrobacterium* colony was inoculated into 5 mL of liquid LB media containing kanamycin (50 μg/mL) for selection and the bacterial culture was incubated at 28 °C for 2 days. The 5 mL of bacterial culture was poured into a flask with 500 mL of liquid LB media containing kanamycin, and the culture was incubated 28 °C for 24 h until it reached the stationary phase in which absorbance at 600 nm (OD600) was between 1.5 and 2.0. Before transferring the bacterial suspension to a 500 mL beaker, 100 μL of Silwet L-77 was added to the bacterial suspension to reduce the surface tension (Silwet L-77 concentration was 0.02%). However, based on our lab experience it was very difficult for the 500 mL of *Agrobacterium* culture to reach OD600 1.5–2.0 even after a 26 h incubation at 28 °C. We modified the protocol; we prepared two sets of tubes, each containing 5 mL of liquid LB media, inoculated with an *Agrobacterium* colony. After incubation at 28 °C for 2 days, the two tubes containing *Agrobacterium* culture were separately poured into two flasks with 500 mL of liquid LB media (5 mL bacterial culture per 500 mL LB). Bacterial pellets collected from the two flasks were combined into 500 mL of 5% sucrose solution.

Four-week-old *A. thaliana* showing 20–30 inflorescences in pots were prepared to be transformed. Siliques were clipped off before the floral dipping process. We inverted the pots and immersed the aerial parts of plants in the 500 mL of the *Agrobacterium* cell suspension no more than 10 s. The dipped plants were covered with plastic bags to maintain the high moisture rate over night. The bags were removed the next day and plants were grown in the growth chamber for a month, then seeds were collected to continue crosses leading to T3 generation plants homozygous for the trans gene(s). Such plants were subsequently tested for segregation of the *trans* gene alleles.

### 2.10. Determination of Copy Number of Trans Genes in A. thaliana

Segregation is a conventional approach to determine whether the transgenic *Arabidopsis* lines are homozygous for the transformed *trans* genes. *A. thaliana* that underwent *Agrobacterium* transformation was considered the T0 generation. The T0 plant yielded seeds which were the T1 progeny. These seeds of T1 progeny were spread onto agar containing MS medium and kanamycin (50 μg/mL). The antibiotic was used to select seedlings successfully transformed with the gene construct *MVLG_05122ΔSP*-CFP, *MVLG_06175ΔSP-mCherry*, and *MVLG_06175-mCherry*. Without the kanamycin resistance gene encoded in the pRI vector, the kanamycin sensitive seedlings usually do not grow well in media and become bleached. In contrast, seedlings homozygous or heterozygous for the *trans* gene and seedlings with multiple insertions of the genetic construct are usually green and flourishing on the media.

The green T1 progeny were transplanted to soil for further growth, and they yielded seeds of T2 progeny. Around 100 seeds of T2 progeny from a single T1 green plant were placed in MS media with kanamycin. Although homozygous and heterozygous *Arabidopsis* strains as well as strains with multiple insertion of gene constructs were all kanamycin resistant, their seedlings would show different ratios in segregation of kanamycin resistance. Since *Arabidopsis* is a species capable of self-pollination, the progeny of a heterozygous strain will have 75% kanamycin-resistant seedlings and 25% kanamycin-sensitive offspring, while a homozygous strain and a multiple-inserted strain will show 100% kanamycin-resistant seedlings. The seedlings showing 100% kanamycin-resistance were discarded, and the 75% kanamycin-resistant green seedlings were selected and transferred to soil for growth. These green seedlings are either homozygous or heterozygous. They yielded seeds of the T3 progeny. Because the T3 offspring batch was due to either heterozygous or homozygous plant strain, the 100% kanamycin-resistant green seedlings of the T3 progeny were considered homozygous and would progress to the subsequent DNA and RNA extraction.

### 2.11. Plant mRNA Extraction

The current study used Zymoclean kit (Zymo Research Corp., Irvine, CA, USA) to conduct RNA extraction of the transgenic *A. thaliana*. Leaves of four-week-old plants were placed in frozen (−80 °C freezer) mortar, and liquid nitrogen was added to quick-freeze leaf tissues. A frozen pestle was used to grind the tissues into fine powder, and the powder was transferred to a 2 mL microcentrifuge tube for extraction using the kit.

### 2.12. qRT-PCR

The eluted plant mRNA was converted to cDNA for use in qRT-PCR. For the synthesis of cDNA, oligo dT primers and Superscript III cDNA synthesis kit (Invitrogen Corp., Waltham, MA, USA) were used. The cycle threshold values of the housekeeping gene *ubiquitin 10* (*UBQ10*) was selected as the standardized index to normalize the RNA expression of the target genes in the study. 1× Power SYBR Green Mango Bio Eva-Green (Applied Biosystems, Waltham, MA, USA) was the detector. The reaction was performed in an Applied Biosystems Step-One thermocycler. Primers were designed to amplify sequences of *MVLG_06175*, *MVLG_05122*, *UBQ10*, *mCherry*, and *CFP*. PCR conditions were 95 °C for 10 min, followed by 95 °C for 15 s, and 60 °C for 1 min. The total process was 40 cycles. Analysis of melting curve was performed at the end of each cycle to ensure the specificity of the reaction.

### 2.13. Fluorescence Confocal Microscopy

Roots of two-week-old Arabidopsis lines expressing MVLG_06175ΔSP-mCherry, MVLG_06175-mCherry, and mCherry alone, leaves of four- or six-week-old Arabidopsis line expressing MVLG_06175ΔSP-mCherry, MVLG_06175-mCherry, MVLG_05122ΔSP-CFP, mCherry alone, and CFP alone, and their corresponding WT controls were observed by confocal microscopy. Images were acquired by an Olympus Fluoview FV-1000 confocal coupled to an Olympus 1 × 81 inverted microscope ((Olympus, East Syracuse, NY, USA), a PlanApoN 60× objective, and FV-10 ASW 2.1 software. A single channel scanning configuration was set up for the acquisition of mCherry (excitation 587 nm, emission 610 nm) and CFP (excitation 458 nm, emission 476 nm) using a 543 nm HeNe laser and a 458 line of argon laser, respectively. Scanning was set at a speed of 2 μs/pixel to acquire z-stacks of each visual field. Images are presented as either single plane images or stacked images.

### 2.14. Fusarium Infection Assay on Arabidopsis thaliana

The growth and infection protocols of Diener & Ausubel [21] were used, with certain modifications. Wild type *A. thaliana* (Col-1, *n* = 67), the transgenic line carrying *MVLG_05122ΔSP-CFP* (*n* = 65), and the transgenic line carrying *CFP* alone (*n* = 63) were infected by *Fusarium oxysporum* strain 5176 (Fo5176) on the 12th day instead of the 2 to 3 weeks as indicated in the original protocol, and the plants were incubated at 29 °C after infection for a potentially more active fungal colonization. The inoculation density was 1.15 × 106 conidia mL^−1^. The current study used the double-blinded experiment design; the identities of plants were not revealed until all disease index scores were collected. The disease index score of symptoms: 5, plants are identical to the wild type plants with the mock inoculation; 4, leafstalks are undersized; 3, the rosette is compact and older leaves showed vascular chlorosis; 2, younger leaves are undersized, and older leaves show chlorosis or yellow color; 1, younger leaves are severely undersized, and older leaves are dead; 0, the plant is completely dead. Results were analyzed using one-way ANOVA in Excel 2016 (Microsoft Corp., Bellevue, WA, USA). This was followed by post hoc analysis with the web-based tool, Tukey HSD (https://astatsa.com/OneWay_Anova_with_TukeyHSD; URL accessed on 5 March 2024).

## 3. Results

### 3.1. Bioinformatic Characterization of Putative Effectors

Protein MVLG_06175 had previously been characterized for prediction as a secreted protein, similarity to other known proteins, and level of intrinsic disorder (a trait associated with fungal effectors) [10]. Additionally, MVLG_05122 and MVLG_02245 were identified in the current study through in silico analysis of the *M. lychnidis-dioicae* genome and transcriptome data. Both possessed features consistent with their categorization as fungal effectors, including up-regulation in the infected host plant. To predict whether the MVLG_02245 protein might similarly play a role in manipulating the plant host of *M. lychnidis-dioicae*, the amino acid sequences for all three species of *Microbotryum* examined in the current study were run against the online effector-prediction software, EffectorP 1.0 [6]. The MVLG_02245 protein was predicted to be an effector in all three species, with a probability score of 0.865, 0.686, 0.645 for *M. lychnidis-dioicae*, *M. silenes-dioicae*, and *M. violaceum* var. *paradoxa*, respectively. To screen for any shared sequence with known effectors, we used the Pfam 32 and HMMER 3.1b1 tools to screen for protein families; however, neither of the programs yielded significant results. The combination of predicted secretion in the sister species pair, the predicted effector function in all three species, and the lack of a Pfam domain indicates that MVLG_02245 is likely an effector with a unique function for the infection of *Silene* hosts. This prediction that MVLG_02245 is a genus-specific effector for *Microbotyrum* is further supported by a lack of blastn and blastp hits to sequence outside the *Microbotryum* genus when screened against the general NCBI genome database. Furthermore, MVLG_02245 is predicted to be a disordered protein (Appendix A). Intrinsically disordered proteins allow for flexibility and have been described as another hallmark for small-secreted effectors for various pathogens [22], including in *Microbotryum* as outlined by Kuppireddy et al. [10].

### 3.2. Yeast Secretion Trap Assay to Verify the Secretion of Effector Proteins

The assay employed the SEY 6210 yeast strain which carries pYSTO-0 vector encoding a leucine gene *LEU2* and an invertase gene *SUC2*. However, the invertase synthesized by the SEY 6210 strain will not be secreted extracellularly because the *SUC2* gene in the pYSTO-0 vector lacks the signal sequence. As a result, the yeast cells will not be able to proliferate in media where sucrose is the sole carbon source and leucine is absent. The signal sequences of *MVLG_02245*, *MVLG_06175* and *MVLG_05122* (see Appendix A) were amplified by PCR and inserted ahead of *SUC2* in-frame in the vector pYSTO-0, and the newly constructed vectors were transformed into yeast strain SEY 6210. The yeasts were cultured in medium in which leucine was absent and sucrose was the only carbon source. If the examined protein effectors are normally secreted from fungal cells, their signal peptides would lead invertase to be secreted out of the yeast cells as well; the yeast could thereby digest sucrose and absorb glucose to grow. This was indeed observed with the YSP test for all three effectors (Figure 1 and Appendix A). Comparing with the yeast transformed with vector only, those transformed with the fungal signal sequence grew well in the media where sucrose was the only carbon source.

### 3.3. Yeast Two-Hybrid Screening to Reveal the Identities of Potential Plant Target Proteins

As indicated above in Section 2.5, yeast strain AH109 transformed with the bait vector containing *M. lychnidis-dioicae* putative effector genes and Y187 transformed with the prey vector were mixed for mating and incubated overnight. The yeast culture was spread and cultivated on quadruple drop-out media lacking adenine, histidine, leucine, and tryptophan (QDO SD/-Ade/-His/-Leu/-Trp). Yeast colonies showing blue color were chosen for repurification on higher stringency QDO medium in which 50 mM 3-AT was added. Such colonies were used for further DNA extraction and sequencing.

However, Y2H is notorious for generating false positive interaction results. The interactions between proteins expressed from the bait vectors and prey vectors observed in Y2H thus require further confirmation by swapping genes into the opposite vectors and repeating the Y2H. That is, the respective effector genes are cloned into the prey vector carried by the yeast strain Y187, and genes of potential targets were cloned into the bait vector carried by the yeast strain AH109.

#### 3.3.1. MVLG_06175 Interacts with Two Fungal Proteins and One Plant Host Protein

The Y2H on MVLG_06175 yielded around 2500 yeast colonies presenting different degrees of blue color on quadruple drop-out media lacking adenine, histidine, leucine, and tryptophan (QDO SD/-Ade/-His/-Leu/-Trp). The media also contained X-α-gal and 3AT (5 mM). Around 1000 colonies were transferred to medium with a more stringent concentration of 3-AT (50 mM) to reduce the leaky activity of the *HIS* allele, and 220 yeast colonies with the deepest blue color were selected for further investigation. Among the 60 samples analyzed by sequencing, 39 were characterized bioinformatically. Comparison of these DNA sequences using blastn or blastx against the database of the Broad Institute and the National Center for Biotechnology Information (NCBI) identified protein products of two *M. lychnidis-dioicae* genes and one plant gene. Of the 39 samples, four matched plant gene SOVF_158740 and/or CASP-like proteins, one matched a *M. lychnidis-dioicae* fungal gene *MVLG_06379*, and the remaining 34 samples matched *M. lychnidis-dioicae* fungal gene *MVLG_07305*. For the matches to plant genes, the highest matches were the SOVF_158740 of spinach (*Spinacia oleracea*), the CASP-like protein 2C1 (CASPL2C1) of beetroot (*Beta vulgaris* subsp. *vulgaris*), the CASP-like protein 2C1 of soybean (*Glycine max*), and CASP-like protein 3 of wild soybean (*Glycine soja*). The CASP-like proteins are homologs of the Casparian strip membrane domain proteins (CASPs) that are associated with the formation of the Casparian strip. CASPs are four-span transmembrane proteins with the carboxyl and amino ends in the cytoplasm. These proteins deposit on the surface of the endodermal cell membrane and polymerize to form a protein scaffold surrounding the endodermis as the precursor of the Casparian strip [23].

In order to confirm the protein-protein interactions were genuine rather than false positives, after initial identification, genes of interest and genes of potential targets were cloned into the opposite vectors to proceed with the Y2H spot test. Additionally, a positive control and certain negative controls were later used in the Y2H spot test as well. MVLG_06175 continued to show interactions with its potential plant protein interactor CASPL2C1 after exchange of vectors. The results indicated that the protein-protein interactions between the phytopathogenic fungus and plant host proteins initially observed were likely genuine (Appendix A).

#### 3.3.2. MVLG_05122 Interacts with Two Plant Host Proteins

A total of 53 blue yeast colonies were acquired on QDO/X-α-gal + 3AT (25 mM) medium from Y2H of MVLG_05122, and 30 of them were analyzed by sequencing. After comparing these DNA sequences using blastx against the database of the National Center for Biotechnology Information (NCBI), the result showed all of the identified protein products of the cDNA library tested matched the constitutive photomorphogenesis 9 (COP9) signalosome complex subunit 5a and/or 5b (CSN5a/5b) from plants. CSN5a and CSN5b are two homologous proteins although the expression level of *CSN5a* is higher in plant cells. The CSN protein complex consists of eight subunits, and CSN5a/5b is the enzymatic center for the isopeptidase activity.

MVLG_05122 also continued to show interaction with the potential plant protein interactors CSN5a/5b after exchange of vectors, suggesting that the protein-protein interactions between the fungus and plant host were genuine as well (Figure 2).

#### 3.3.3. MVLG_02245 Interacts with Four Plant Host Proteins

Plasmid extraction followed by sequencing for 50 diploid colonies obtained on QDO dropout medium demonstrated that four interactions were found more predominantly than others. These plant protein sequences were compared against the NCBI database and yielded plant orthologs for a ferredoxin-thioredoxin reductase catalytic chain protein, a Photosystem II protein, a xyloglucan endotransglucosylase/hydrolase 4 protein, and a Tubulin α-1 chain protein (tα-1c). Only the fourth interaction, MVLG_02245 X tα-1c, yielded blue colonies when the coding regions were swapped between the bait and prey vectors and mating was repeated (Figure 3). While the interaction between MVLG_02245 and tα-1c was confirmed in the vector swap, the resulting diploids grew slower in both the DDO and QDO media (Figure 3).

### 3.4. Potential Locations of Protein-Protein Interactions in the Plant Host

To reveal the subcellular locations of effector proteins in the heterologous model plant (i.e., *A. thaliana*) tissues, the fungal genes MVLG_*06175* and *MVLG_05122* were linked to fluorescent tag genes expressing mCherry (*MVLG_06175-mCherry* and *MVLG_06175*Δ*SP-mCherry*) or the cyan fluorescent protein (CFP) (*MVLG_05122*Δ*SP-CFP*), respectively, through Gibson assembly [18]. These gene constructs, whose transcription was driven by CaMV 35S promoter, were transformed into *A. thaliana* by *Agrobacterium*-mediated transformation. Segregation analysis confirmed the genetically modified *A. thaliana* were homozygous for the transgenes in the third generation (T3 progenies). mRNA of the transgenic plants was extracted to make cDNA, followed by qRT-PCR to confirm the expression of the gene constructs.

Since *Arabidopsis thaliana* served as the model plant to investigate the effects of expression of the fungal effector proteins, it was necessary to determine whether the protein-protein interactions occurred *in planta* as well. *A. thaliana* also synthesizes CSN5a/5b and CASPL2C1 proteins, but the amino acid sequences of these proteins in *Silene latifolia* are not identical to those in *A. thaliana*. CSN5a and 5b in *A. thaliana* and *S. latifolia* share 82.4% and 81.1% identity in amino acid sequence, respectively, while CASPL2C1 in the two plants shares only 48.7% identity at the amino acid level. Primers were designed to acquire *CSN5a/5b* and *CASPL2C1* from *A. thaliana* genome, and the plant genes were inserted into the prey vector to be used in Y2H spot tests. The outcomes showed the interactions between MVLG_5122 and CSN5a of *A. thaliana* gave a positive result on QDO dropout media, but MVLG_06175 did not provide such a result with the CASPL2C1 ortholog of *A. thaliana*. (Appendix A).

#### 3.4.1. Localization of Effector Protein MVLG_06175 in Transgenic Plant Tissues

In the four-week-old transgenic *A. thaliana*, the signals of mCherry linked to MVLG_06175ΔSP and MVLG_06175 displayed granules clustered at the tips of trichomes on leaves. In contrast, signals of mCherry alone expressed in the plant as the positive control concentrated at the center of the trichomes, and auto-fluorescent signals of the wild type *Col*-0 (WT) displayed a weaker and relatively random distribution (Figure 4). Since CASPL2C1 could be involved in the formation of the Casparian strip, fluorescence images of the root cap were taken. Transgenic *A. thaliana* strains expressing *MVLG_06175ΔSP-mCherry*, *MVLG_06175-mCherry*, and *mCherry* alone exhibited clear mCherry signals in the roots of the two-week-old seedlings in comparing with the WT plant. While plants carrying *mCherry* alone did not show specific patterns, those carrying *MVLG_06175ΔSP-mCherry* displayed concentrated granules, and those carrying *MVLG_06175-mCherry* either formed granules with a much weaker fluorescent intensity or showed no specific patterns (Figure 5). The punctated signals located at the tips of the leaf trichomes and the root cap indicated potential interactions between MVLG_06175 and CASPL2C1 or other host protein(s) in those locations.

#### 3.4.2. Localization of Effector Protein MVLG_05122 in Transgenic Plant Tissues

Confocal fluorescence images of *MVLG_05122ΔSP-CFP* transgenic *A. thaliana* showed similar overall intensity of signals to those of the *CFP* alone transgenic line and those of wild type *Col*-0 (WT). However, there were significant and clear band structures at the bases of trichomes on leaves of the *MVLG_05122ΔSP-CFP* transgenic plant. The band-like CFP signals were also observed in trichomes on leaves of WT and *CFP* alone transgenic lines, but the *MVLG_05122ΔSP-CFP* transgenic line displayed the strongest signal intensity (Figure 6).

### 3.5. Determination of Phenotype Changes in Transgenic A. thaliana

We examined four parameters of *A. thaliana* plants: leaf quantity, rosette diameter, days to flower opening, and silique quantity in the T3 progeny of transgenic *A. thaliana.* Based on segregation analyses, these plants are homozygous lines (see Section 2.10). Plants were initially grown in MS media and transferred to soil at 14 days. All parameters showed no differences except the rosette diameter and leaf quantity in the four-week-old plants transformed with MVLG_06175. Both *MVLG_06175-mCherry* and *MVLG_06175ΔSP-mCherry* transformed *A. thaliana* had statistically smaller rosette diameter and leaf quantity than the *mCherry* alone transformed plants and WT (Appendix A). Data were collected from 34 plants of *MVLG_06175ΔSP-mCherry*, 28 plants of *MVLG_06175-mCherry*, 35 plants of *mCherry* alone, and 35 plants of the wild type *(Col-0*). In contrast, there were no statistically significant differences in the four parameters among the *MVLG_05122ΔSP-CFP* trangenic line, *CFP* alone trangenic line, and WT.

### 3.6. Fusarium Infection Assays on Transgenic A. thaliana

A fungal infection assay was conducted with *Fusarium oxysporum* strain 5176 (Fo5176) to determine whether constitutive expression of fungal effector proteins in the transgenic *Arabidopsis* lines would change plant immune responses against phytopathogenic infection. Since the plant protein CSN5a of *A. thaliana* showed interactions with MVLG_05122 in Y2H (Appendix A), transgenic *A. thaliana* line expressing MVLG_05122-CFP was selected to undergo the infection assay. The result of the infection assay showed statistically significant differences among the *A. thaliana* line expressing MVLG_05122ΔSP-CFP, the line expressing CFP alone, and expressing the wild type line; *A. thaliana* line expressing MVLG_05122ΔSP-CFP demonstrated higher disease index scores than the other two control groups (Figure 7). Data were collected from 65 plants of *MVLG_05122ΔSP-CFP*, 63 plants of *CFP* alone, and 67 plants of the wild type (*Col-0*).

## 4. Discussion

The current study examined one previously identified effector of *M. lychnidis-dioicae*, MVLG_06175. In addition, here we present data on two other putative effectors, MVLG_05122 and MVLG_02245. All three were identified initially via bioinformatic approaches, as well as transcriptome data showing up-regulation *in planta*. Beckerson et al. [24] compared secretomes for several species in the *Microbotryum* genus and found that host-specialization in the genus is likely due to conserved but rapidly evolving shared sets of effectors. Of the secreted effectors identified, many had orthologous non-secreted variants in other species. Therefore, host-specialization in the *Microbotryum* complex may be driven not only by stepwise changes to core secreted proteins, but may also be driven by the mobilization of effectors through changes to the signal peptide region of the gene. The preservation of the coding sequence for MVLG_02245, together with the changes observed in the signal peptide region for different *Microbotryum* species, makes this particular putative effector an interesting candidate for functional analysis. We here applied yeast two-hybrid screening (Y2H) to identify plant proteins of *Silene latifolia* as potential targets for the three fungal effectors of study. MVLG_02245 was found to interact predominantly with four plant proteins, including the one for Tubulin α-1 chain protein (tα-1c). Additionally, the Casparian strip membrane domain-like protein 2C1 (CASPL2C1) and COP9 signalosome subunit 5a/5b (CSN5a/5b) of the plant host were found to interact with MVLG_06175 and MVLG_05122, respectively. To further elucidate the influences of fungal proteins MVLG_06175 and MVLG_05122 on host plants during fungal colonization, these latter two genes were stably expressed in the model plant *A. thaliana*. In order to examine changes *in planta* potentially resulting from the protein-protein interactions, and to identify the subcellular locations of the interactions in the plant cells, MVLG_06175 and MVLG_05122 were linked to mCherry and cyan fluorescent protein (CFP), respectively (MVLG_06175ΔSP-mCherry, MVLG_06175-mCherry, and MVLG_05122ΔSP-CFP).

### 4.1. MVLG_06175 Could Affect the Formation of the Casparian Strip and Growth of Trichome by Interacting with CASPL2C1

Casparian-strip membrane domain proteins (CASPs) form a protein scaffold for lignin polymerization during the synthesis of the Casparian strip [23]. The Casparian strip is a specialized cell-wall structure surrounding the endodermis. It separates the cortex and stele as well as regulates the flow of water and transport of solutes in the plant. It blocks apoplastic diffusion so that all solutes, salts, and water can only enter the xylem and phloem through the cytoplasm of the endodermis [23,25,26,27]. The Casparian strip mainly forms in the root, but it also is found in specific tissues of certain plant species, such as the stem of pea [28], pine needles [29], and the leaves of quillworts [30]. The Casparian strip might be directly involved in plant defense against microbial invasion as well. Interestingly, it has been demonstrated that the hyphae of mycorrhizal fungi, forming a mutualistic relationship with plants, are unable to pass the Casparian strip to enter the stele of roots [31]. Both CASPs and CASP-like proteins (CASPLs) are integral membrane proteins with four transmembrane helices. This type of protein structure is a common feature in the myelin and lymphocyte (MAL) domain family and is related to proteins for vesicle trafficking and membrane link (MARVEL) domain protein family in the animal kingdom. MARVEL domain proteins are associated with the functions of epithelial tight junctions [32]. Since CASPLs and MARVEL domain proteins are orthologous, CASPLs could as well be involved in tight-junction functions in plant cells. Relating this information to fungal infection, *M. lychnidis-dioicae* might alter the formation or structure of the Casparian strip by the interactions between MVLG_06175 and CASPL2C1; facilitated by the fungal effector protein, the invasive hyphae could then be able to penetrate into the xylem and phloem of the plant host. The hyphae of *M. lychnidis-dioicae* are observed in the intercellular space in the rootstocks of *S. latifolia* [33]. Alterations of the Casparian strip resulting from fungal effectors might also occur at the meristem of new tissues such as shoots.

CASPs are expressed exclusively in the endodermal cells for the polymerization of lignin, but CASPLs can be expressed in different tissues such as abscission zone cells, peripheral root cap cells, trichomes, and xylem pole pericycle cells, where they could function as modifiers of cell-wall-related structures [34]. As homologs of CASPs, CASPLs might form a protein scaffold for lignin deposition to form the Casparian strip, too. However, the polymerization of lignin on the protein scaffold guided by CASPLs could also occur at the pathogenic infection sites in order to isolate microorganisms. Fluorescence-labeled CASPL4D1 (a homolog of CASP4) and CASPL1D1 (a homolog of CASP1) were found to gather and encircled the bacterial plant infiltration sites of *Pseudomonas syringae*. Disruption *CASPL4D1* and *CASPL1D1* in *A. thaliana* significantly increased colony forming units of *P. syringae*. Such results suggest that the deposition of lignin on the protein scaffold could be applied to isolate the pathogen at the infection sites [35]. From the point of view of phytopathogens, mitigating the functions of CASPL4D1 and CASPL1D1 by effector proteins, if they exist, would help the spreading of pathogens in *A. thaliana*.

Although the in vitro Y2H spot test (Appendix A) did not detect the interactions between MVLG_06175 and CASPL2C1 of *A. thaliana*, fluorescence-labeled MVLG_06175 showed different signal distributions in the transgenic *A. thaliana* in comparison with the WT and mCherry alone transgenic plants. The images revealed the localization of MVLG_06175 at the tip of trichomes and of roots. We expected that the MVLG_06175 without signal sequence would stay in the cytoplasm, while MVLG_06175 with signal sequence would be secreted out of the cell, but our images showed their fluorescence signals exhibited a similar pattern in the trichomes; they all aggregated at tips of trichomes, like small particles (Figure 4). In terms of mCherry signals in the roots, MVLG_06175 without signal sequence showed aggregation at the root tips, but MVLG_06175 with signal sequence did not or the aggregation was very weak (Figure 5). Considering the in vitro Y2H spot test, the aggregation of fluorescence signals in trichomes and roots might indicate that MVLG_06175 was interacting with CASPL2C1 of *A. thaliana* in the two tissues, despite the fact that in vitro Y2H did not show the interactions (Appendix A) in the two tissues.

The fluorescence images only indicated where the fungal effector proteins were located in the plant. The true identities of *A. thaliana* plant proteins interacting with the effector proteins in the transgenic plants may require a co-immunoprecipitation assay to reveal. Nevertheless, CASPL2C1 still could be a candidate, and the smaller rosette diameter and leaf quantity observed in *MVLG_06175* transgenic *A. thaliana* (Appendix A) might result from the modifications of plant proteins by the fungal effector protein. Although there were no studies directly tracing the distribution of CASPL2C1 in plant tissues, a former study showed the fluorescence-labeled CASPL5B1 of *A. thaliana*, a homologous protein of CASP5, was expressed in immature and differentiated trichomes [34]. Therefore, we cannot exclude the possibility that the MVLG_06175 is interacting with other CASP-like proteins in trichomes. As the first defensive line of the plant, trichomes are the protruding epidermal structures on aerial plant tissues. They protect plants from ultraviolet (UV) light damage, insect bites, excess water transpiration, and drastic temperature changes [36]. Phytochemicals synthesized and secreted from trichomes are often involved in resistance to phytopathogens as well as pollinator attraction [37]. If the interactions between the effector protein MVLG_06175 and plant protein CASPL2C1 do happen at the tips of trichomes, the protein-protein interaction could be involved in the trichome development and flavonoid synthesis, and *M. lychnidis-dioicae* might be neutralizing the immune responses of *S. latifolia* to enhance the colonization. Since *M. lychnidis-dioicae* starts its infection cycle on the flowers of the host plants or in wounded host tissues, the modifications potentially happening in trichomes on the flower buds could provide advantages to the fungal invasion.

### 4.2. MVLG_05122 Might Alter the Protein Turnover Rate Associated with the Plant Defense and Trichome Growth by Interacting with the Plant Proteins CSN5a/5b

Y2H and sequencing results showed that the plant proteins constitutive photomorphogenesis 9 (COP9) signalosome complex subunit 5a and 5b (CSN5a/5b) potentially interacted with the fungal effector protein MVLG_05122. The CSN protein complex is composed of eight subunits, and subunits 5a and 5b are the catalytic center with the isopeptidase activity. The two subunits are homologous proteins in *S. latifolia*, with *CSN5a* being more highly expressed and mutations in that subunit producing more prominent phenotypes [38]. In the model plant, *A. thaliana*, the homozygous silencing or knockdown of *CSN5b* did not cause significant changes in the phenotype of the plant. In contrast, homozygous silencing or knockdown of *CSN5a* resulted in significant growth defects such as reduced growth at seedling and adult stages, impaired lateral root formation, impaired root hair formation, loss of apical dominance, depleted trichomes, and smaller flower size. It is noteworthy that silencing of both *CSN5a* and *5b* led to lethality at the seedling stage [39,40].

The CSN protein complex could interfere with the enzymatic activity of Cullin-RING E3 ubiquitin ligases (CRLs), a superfamily of ubiquitin ligases that transfers ubiquitin to proteins that are subjected to degradation by the 26S-proteasome [41]. CSN5a/5b functions as an isopeptidase, catalyzing the removal of the Nedd8 protein from CRLs. Without the subunit, Nedd8 the CRLs are dissociated and unable to transfer ubiquitin to the protein destined to be degraded [42]. The ubiquitination and degradation of proteins through CRLs and 26S-proteasomes are important in regulating plant development and immune responses such as the induction of genes, oxidative burst, hormone signaling, and programmed cell death (PCD). PCD is a type of plant cell autophagy where host plants sacrifice some tissues to prevent phytopathogens from invasion and spread. [43,44]. Plants have evolved a large quantity of proteins to adjust rates of protein turnover. Taking *A. thaliana* as an example, up to 6% of the plant proteome is associated with protein removal, and the plant genome encodes more than 1400 different E3 ligase components [45]. Due to their prominent influences, these plant ligases and proteasomes have become potential targets to be modified by various phytopathogen species. Phytopathogenic effector proteins could directly bind to the CRLs to alter the rate of ubiquitination. AVR3a is an effector protein synthesized by *Phytophthora infestans*, an oomycete species infecting potato, maize, and tobacco. The U-box E3 ligase CMPG1 of the plant host constantly undergoes self-ubiquitination and the subsequent degradation by the 26S-proteasome, and the fast degradation of the CMPG1 is essential to initiate programmed cell death (PCD) of plant tissues. The binding of AVR3a to CMPG1 stabilizes CMPG1 by preventing further self-ubiquitination. The reduction in degradation rate of the CMPG1 might in turn decrease the occurrence of PCD during the colonization of *P. infestans* [46]. Additionally, the effector proteins may function as a ubiquitin ligase rather than bind to host E3 ligases. The type III secretion system effector AvrPtoB of *Pseudomonas syringae* pv. *tomato*, a bacterial species infecting tomato, carries a domain mimicking the activity of plant E3 ubiquitin ligases. The bacterial effector protein could ubiquitinate many plant host kinases responsible for the initiation of immune responses. The higher degradation rate of these kinases leads to increased susceptibility of tomatoes to the bacterium [47]. Effector proteins could also target the proteasome to alter the protein degradation rate. Syringolin A (SylA) synthesized and secreted by *P. syringae pv syringae* is a virulence factor covalently binding to the catalytic subunits of eukaryotic 20S proteasomes, which causes irreversible inhibition of the proteasomes [48]. The *sylA* mutated bacteria are unable to reduce the accumulation of salicylic acid, leading to the activation of the PCD in host plants [49].

With respect to MVLG_05122 of *M. lychnidis-dioicae*, perhaps it modifies the ubiquitination and degradation rate of plant proteins, as well, specifically those involved in plant immunity. To review such responses, as phytopathogens enter plant host tissues, the plant disease resistance (R) receptor proteins recognize and bind to effector proteins released from pathogens. The perception of effector proteins initiates effector-triggered immunity (ETI). The immune responses include elevated salicylic acid (SA) concentrations and PCD to limit further colonization by the phytopathogens [50]. These responses are mediated by a transcriptional coactivator NPR1. NPR1 also triggers the expression of genes associated with systematic acquired resistance (SAR). SAR grants plants immunity against a broad spectrum of phytopathogens for a longer period of time. NPR1 forms oligomers in the cytoplasm by disulfide bonds between two cysteine residues. As the microbial infection progresses, the cellular SA concentration continues rising, leading to changes in the cellular oxidation-reduction status. The disulfide bonds within the NPR1 oligomers are broken due to the redox changes, and oligomers turn into monomers which translocate to the nucleus to stimulate defense-related gene expression. It is estimated that this transcriptional coactivator regulates expression of more than 2200 genes in *A. thaliana* [51,52,53].

Cytoplasmic levels of R receptor proteins and the following signaling pathways are tightly regulated to avoid autoimmune responses in the plant. The Cullin 1-RING E3 ubiquitin ligase (CRL1) consisting of the CUL1 backbone is one of the ubiquitin ligases responsible for ubiquitination of R receptor proteins for their subsequent degradation by 26S-proteasome [51]. MVLG_05122 potentially interacts with CSN5a/5b, which normally catalyzes the cleavage of Nedd8 protein from CRLs. The removal of Nedd8 inhibits the CRLs enzymatic activity and thus hinders the CRLs-mediated ubiquitination on R receptor proteins [42]. Furthermore, the interactions between MVLG_05122 and CSN5a/5b might also affect the activity of the Cullin 3-RING E3 ubiquitin ligase (CRL3), another member of the E3 ligase family. CRL3, consisting of the CUL3 backbone, can alter cellular levels of NPR1 by ubiquitination and the following protein degradation [51]. By interacting with the plant protein complex CSN, which is responsible for regulation of CRL3, the MVLG_05122 could affect the cytoplasmic concentrations of NPR1.

If MVLG_05122 does interact with CSN, it is not clear whether this effector would inhibit or enhance the CSN protein complex activity. However, the result of fungal infection assay (Figure 7) showed that the transgenic *A. thaliana* line expressing MVLG_05122 exhibited better resistance against the infection of *F. oxysporum* strain 5176, suggesting that the protein-protein interactions could actually mitigate fungal infection. This is opposite to our prediction since heterologous expression of fungal effector protein MVLG_05122, absent the fungus in these transgenic *A. thaliana* lines, reduces the fungal colonization of the plant host. If this is true, the protein-protein interactions between MVLG_05122 and CSN5a/5b could intensify the enzyme activities of CSN protein complex to further dismantle both CRL1 and CRL3 (Figure 8); thus, the ubiquitination and degradation rates of the R receptor and NPR1 transcriptional coactivator would be reduced. As a result, the cellular concentrations of the two plant proteins might be increased, potentially strengthening the plant immune responses. Another possible explanation is that the overexpression of MVLG_05122 effector protein could stimulate the plant immune responses against fungi; the plant cells recognize the presence of effector protein and thus the ETI is constantly active, stimulating plant immunity. The plant hormone, jasmonic acid (JA), is involved in mounting a host defense against necrotrophic fungi. The fungus used in the current study against the *A. thaliana* lines, *F. oxysporum*, is known to convert from hemibiotrophic to necrotrophic lifestyle by 6 dpi [52]. This timeframe coincides with up-regulation of JA-related genes in the roots in response to the necrotrophic fungal infection [52]. Perhaps this is an additional explanation for the protective effect of MVLG_05122 expression, serving to prime the plants for the infection of *F. oxysporum.* This is in contrast to what we expect its role could play during a biotrophic infection of a native host plant (i.e., *S. latifolia*) by *M. lychnidis-dioicae*.

It is even more intriguing that the potential interactions between MVLG_05122 and CSN5a also appeared in the trichome. The CFP signals formed an organized band at the basal cells of trichomes (Figure 6), suggesting that the effector protein MVLG_05122 might be affecting trichome growth through interactions with the CSN protein complex. CSN5a has previously shown influences on trichome development and metabolism. Inactivation of *CSN5a* in *A. thaliana* results in elevated production of many carotenoids and phenylpropanoids including anthocyanins in seeds and leaves. This mutated plant line also had significantly reduced density and abnormal morphology of trichomes. These phenotypic changes could be due to altered gene expressions in two tri-protein complexes when *CSN5a* is disrupted [54], although the same changes could also have resulted from altered hormone signaling pathways resulting from inactivation of *CSN5a*. Plant hormones, including JA, salicylic acid (SA), and gibberellin regulate the growth of trichomes [55]. JA, in conjunction with gibberellin induces trichome production, while SA decreases the number and density of trichomes [55]. An earlier study silencing the *CSN5a* in tomato found reduced synthesis of JA but not of SA [56]. A lower JA concentration is a potential cause of the reduced density and abnormal morphology of trichomes, too.

It is as striking a possibility that, during infection, *M. lychnidis-dioicae* might modify the trichome structure and metabolism, and it is more interesting that the fungus might utilize two different effector proteins to interfere with the development of trichomes by interacting with two different plant proteins, CASPLs and CSN5a. These findings suggest that *M. lychnidis-dioicae* may have evolved to apply multiple effector proteins to modulate the same plant structures and/or metabolic pathways.

## 5. Conclusions

This study identified, via Y2H, potential host protein targets for three putative effectors of *M. lychnidis-dioicae*. The predicted interaction of MVLG_06175 with host CASPs or CASPLs, combined with observed localization of MVLG_06175 at root tips and in trichomes of transgenic *A*. *thaliana*, is consistent with a role of the effector in altering host-cell permeability and possible susceptibility to fungal infection. Similarly, the proposed target of MVLG_05122, CSN5a/5b, as well as localization of MVLG_05122 to the base of trichomes in transgenic *A. thaliana*, again suggest a role in host susceptibility to infection; this is borne out by the apparent triggering of ETI when MVLG_05122 was constitutively expressed in transgenic *A. thaliana*, absent the fungus. Finally, MVLG_02245 may represent the mobilization of a core effector through changes to the signal peptide region of the gene.

## Figures and Tables

**Figure 1 jof-10-00262-f001:**
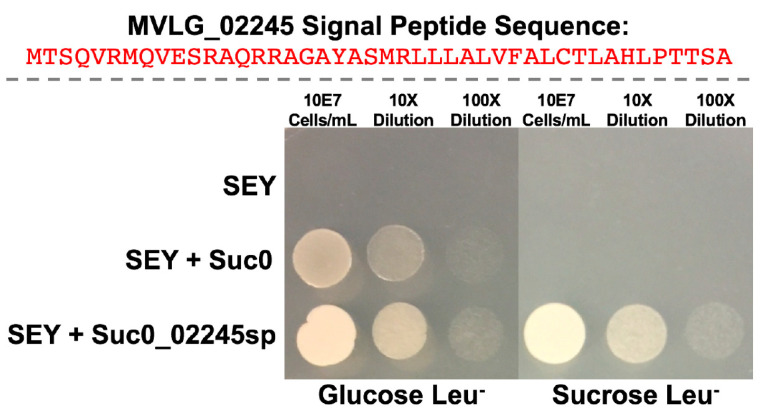
Yeast Secretion Trap results for the signal peptide region of MVLG_02245, predicted using SignalP4.1, after 2 days of growth on Glucose Leucine Dropout Media, left, and Sucrose Leucine Dropout Media, right. In the top row are untransformed SEY strain cells of *Saccharomyces cerevisiae*. The second row contains SEY cells transformed with just the Suc0 vectors. The third row contains SEY cells transformed with the Suc0 vector containing the signal peptide from MVLG_02245 cloned upstream and in-frame of the region encoding invertase enzyme. Similar results for MVLG_05122 and MVLG_06175 are shown in Appendix A.

**Figure 2 jof-10-00262-f002:**
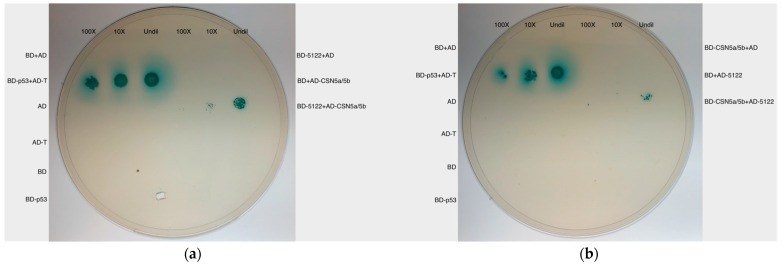
Y2H spot test. Y2H spot tests with and without vector-switch were conducted to reconfirm the protein-protein interactions. (**a**) The result also showed MVLG_05122 interacted with CSN5a/5b. (**b**) The protein-protein interactions remained after the switch of the vector containing the fungal and plant genes. BD-p53+AD-T, a positive control for interaction; AD, AD-T, BD, and BD-p53, the negative controls; BD-5122+AD, BD+AD-CSN5a/5b, BD-CSN5a/5b+AD, and BD + AD-5122, one of the two mating yeast strains carries bait or prey vectors with no insertions, as negative controls; Undil, undiluted; 10× and 100×, 10-fold and 100-fold of dilutions.

**Figure 3 jof-10-00262-f003:**
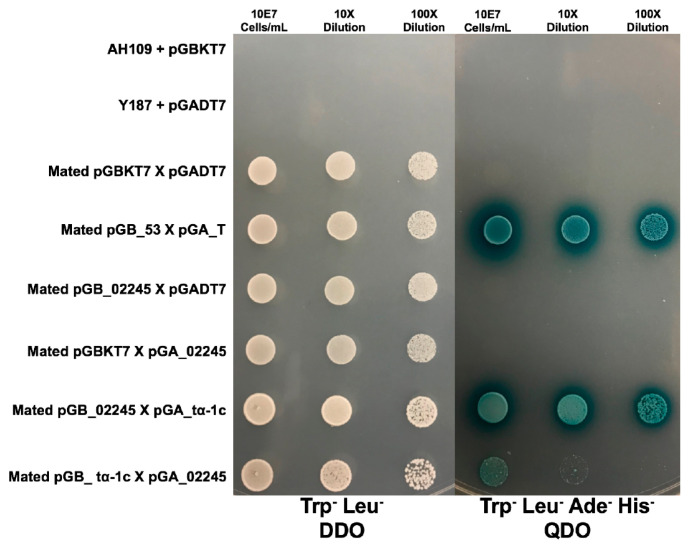
Yeast two-hybrid mating results between MVLG_02245 and tα-1c after 2 days of growth on DDO, left; and 4 days of growth on QDO, right. A series of negative controls were used including the AH109 yeast strain transformed with an empty bait vector (pGBKT7), top row; the Y187 strain transformed with an empty prey vector (pGADT7), second row; Diploid offspring of mated strains containing both the empty bait and empty prey vectors, third row; Diploid cells containing the MVLG_02245 bait vector and the empty prey vector, fifth row; and Diploid cells containing the empty bait vector with the tα-1c prey vector, sixth row. Diploid cells containing the bait and prey vectors for known strong interactors p53 and T-antigen were used as a positive control in the fourth row. Diploid cells containing the MVLG_02245 bait vector and tα-1c prey vector are spotted in the seventh row, and diploid cells containing the swapped tα-1c bait vector and MVLG_02245 prey vector are spotted in the seventh row.

**Figure 4 jof-10-00262-f004:**
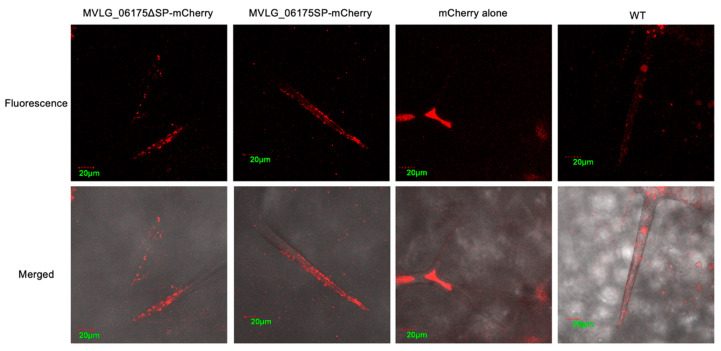
Localization studies of MVLG_06175 in trichomes in 4-week-old *A. thaliana*. Confocal fluorescence images were taken from leaves of the 4-week-old stable transgenic *A. thaliana* expressing *MVLG_06175-mCherry*, *MVLG_06175ΔSP-mCherry*, and *mCherry* alone (Red color in all cases), as well as of the WT plant. Signals of *MVLG_06175-mCherry* and *MVLG_06175ΔSP-mCherry* transgenic lines formed granules clustered at the tips of trichomes on leaves. Expression was under the control of CaMV 35S promoter. In each sample, the upper panel is the fluorescence image and the lower panel is the merged image. Size bar, 20 mm.

**Figure 5 jof-10-00262-f005:**
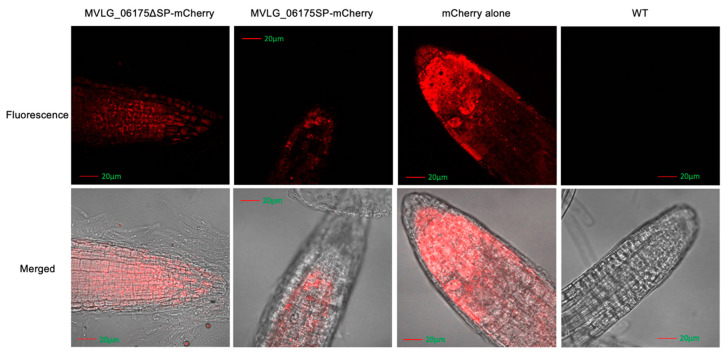
Localization studies of MVLG_06175 in roots in 2-week-old *A. thaliana*. Confocal fluorescence images were taken from roots of the 2-week-old stable transgenic *A. thaliana* expressing *MVLG_06175-mCherry*, *MVLG_06175ΔSP-mCherry*, and *mCherry* alone (Red color in all cases), as well as of the WT plant. Signals of *MVLG_06175ΔSP-mCherry* transgenic lines formed granules in the roots, but those of *MVLG_06175ΔSP-mCherry* transgenic lines did not or formed granules with a weak intensity. Expression was under the control of CaMV 35S promoter. The upper panel is the fluorescence image and the lower panel is the merged image. Size bar, 20 mm.

**Figure 6 jof-10-00262-f006:**
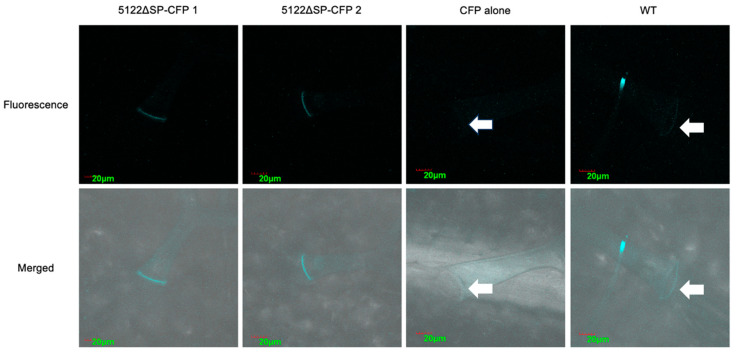
Fluorescence images of trichomes in WT and transgenic *A. thaliana* expressing MVLG_05122ΔSP-CFP and CFP alone. Images of the *MVLG_05122ΔSP-CFP* transgenic line were from leaves of two plants. The band-like CFP signals were stronger in the *A. thaliana* expressing MVLG_05122ΔSP-CFP. Arrows indicate the barely seen band-like signals in the CFP alone and WT. Expression of *trans* genes was driven the by constitutive CaMV 35S promoter. Size bar, 20 μm. The upper panel is the fluorescence image and the lower panel is the merged image. Size bar, 20 μm. Arrows indicate faint or autofluorescence for CFP alone and WT images.

**Figure 7 jof-10-00262-f007:**
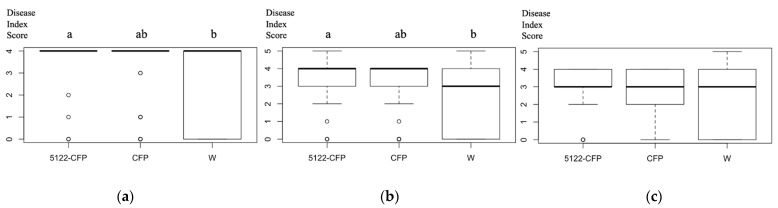
Disease index scores of three different *A. thaliana* lines infected by Fo5176 at (**a**) 10 days, (**b**) 16 days, and (**c**) 22 days post inoculation (dpi). One-way ANOVA analysis and Tukey HSD show that the plant line expressing MVLG_05122 without the signal sequence is significantly higher in the disease index score (i.e., is more resistant to disease symptoms) than the wild type plant at 10 days and 16 dpi (both *p* < 0.01, a and b indicate the statistical differences between the two groups). However, the *p*-value for the 22 dpi group, was just at the boundary of statistical significance. 5122-CFP, *A. thaliana* expressing *MVLG_05122ΔSP-CFP*, *n* = 65. CFP, *A. thaliana* expressing the cyan fluorescence gene, *n* = 63. W, the Col-1 *A. thaliana* as the wild type, *n* = 67.

**Figure 8 jof-10-00262-f008:**
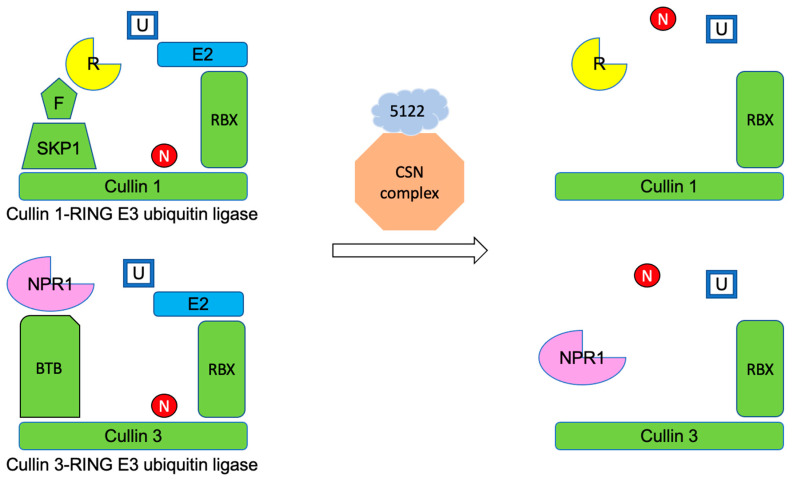
A hypothetical model of the increased enzymatic activity of CSN protein complex resulting from the interactions between MVLG_05122 and CSN5a/5b. The removal of Nedd8 protein, which is catalyzed by the CSN protein complex, causes the dissociation of the Cullin 1-RING E3 ubiquitin ligase (CRL1) and Cullin 3-RING E3 ubiquitin ligase (CRL3), which in turn inhibits the ubiquitination of the R receptor protein and NPR1 transcriptional coactivator. When MVLG_05122 is interacting with the CSN5a/5b, which is the catalytic center of the protein complex, the dissociation of both E3 ligases might be further enhanced. U, ubiquitin; E2, E2 enzyme conjugating ubiquitin; RBX, RING BOX-1 protein; N, nedd8 protein; F (F-box protein), SKP1 (S-phase kinase-associated protein 1), and BTB (bric-a-brac, tramtrack, and broad complex), adaptor and receptor proteins recognizing R receptor protein and NPR1 transcriptional coactivator.

## Data Availability

The annotated *M. lychnidis-dioicae* genome is available at https://mycocosm.jgi.doe.gov/Micld1/Micld1.home.html. (URL accessed on 4 April 2016, 9 September 2019) Transcriptome data are available at NCBI as SRA, at https://www.ncbi.nlm.nih.gov/sra/SRX1819603[accn] (URL accessed on 13 March 2024).

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
