# Peer review of "Characterization of Microbotryum lychnidis-dioicae Secreted Effector Proteins, Their Potential Host Targets, and Localization in a Heterologous Host Plant"

_jof, 2024, doi:10.3390/jof10040262_

Round 1

Reviewer 1 Report

The manuscript submitted by Tsai et al. describes the possibilities that three effector candidates of Microbotryum lychnidis-dioicae might interact with their associated plant proteins. Although the authors showed potential interactions and localizations, I feel the data provided here are not adequate to conclude the current title. My major concern is that the authors showed localization of the effector proteins and performed infection assay only using a heterologous system in Arabidopsis thaliana, which is not an authentic host for this fungus. 

1.     The title should be modified to be appropriate to the content. The authors did not demonstrate that the effectors indeed “manipulate its host plant”. This title can lead to misreading.

2.     Compared to the experiment using MVLG_06175, I could not understand why the authors used only MVLG_05122 “without signal sequence” but not “with the signal sequence”. Since it is still not known whether this effector candidate is a cytoplasmic or apoplastic effector, the authors should provide the date with the signal sequence.

3.     I am not sure whether the median line shown in Figure 7 is correctly drawn or not. Please check it.

4.     Since the experiments were performed only in the heterologous system, the authors should discuss how likely this observation could be reproducible in the host plant, Silene latifolia.

Author Response

Major comments

The manuscript submitted by Tsai et al. describes the possibilities that three effector candidates of Microbotryum lychnidis-dioicae might interact with their associated plant proteins. Although the authors showed potential interactions and localizations, I feel the data provided here are not adequate to conclude the current title. My major concern is that the authors showed localization of the effector proteins and performed infection assay only using a heterologous system in Arabidopsis thaliana, which is not an authentic host for this fungus. 

We appreciate the Reviewer's comments and suggestions. However, we do not agree that expression of an effector in a heterologous host system is invalid. As explained below, such studies have been conducted in other fungal/plant systems, especially where the native host plant cannot be easily genetically manipulated, as is the case with S. latifolia.

Detail comments

1. The title should be modified to be appropriate to the content. The authors did not demonstrate that the effectors indeed “manipulate its host plant”. This title can lead to misreading.

Answer: The title has now been changed to: Characterization of Microbotryum lychnidis-dioicae Secreted Effector Proteins, their Potential Host Targets, and Localization in a Heterologous Host Plant

2. Compared to the experiment using MVLG_06175, I could not understand why the authors used only MVLG_05122 “without signal sequence” but not “with the signal sequence”. Since it is still not known whether this effector candidate is a cytoplasmic or apoplastic effector, the authors should provide the date with the signal sequence.

Answer: Based on the predicted  amino acid sequence of MVLG_05122, the protein is low in Cys residues (1/173; https://mycocosm.jgi.doe.gov/cgi-bin/dispTranscript?db=Micld1&id=6340&useCoords=1&withTranslation=1&dispRuler=1) and so is predicted to be cytoplasmic in  the host plant ( Sperscheiider and Dodds, 2022; https://doi.org/10.1094/MPMI-08-21-0201-R). For this reason, transgenic plants were not generated that contained the secretion signal for this protein.

3. I am not sure whether the median line shown in Figure 7 is correctly drawn or not. Please check it.

Answer: We have re-checked the figure, and actually have re-drawn it to better reflect the plot and the significant differences observed.

4. Since the experiments were performed only in the heterologous system, the authors should discuss how likely this observation could be reproducible in the host plant, Silene latifolia.

Answer: Heterologous expression of putative effectors is not an uncommon method of discerning possible functional roles and localization in planta, absent the fungus (for example, see, Ghareeb et al., 2015. DOI: 10.1104/pp.15.01347 ). This is particularly the case for evaluating function of effectors in host plants that are not easily amenable to genetic manipulation, as is the case with Silene latifolia. Moreover, observations in such transgenic systems have been used to predict functional aspects of effectors in their native hosts. Additional support for our use here is the finding that the targets identified for MVLG_05122, CSN5a and 5b,  share 82.4% and 81.1% identity in amino acid sequence, respectively, between A. thaliana and S. latifolia.

Reviewer 2 Report

Dear authors,

The manuscript is well written and has some interesting data concerning new effectors and their host targets form M. lychnidis-dioicae. The research is well done with all the appropriate controls.

The use of the artificial host A. thaliana and the results for the effector MVLG_06175 make the interpretation of the putative host target protein a bit difficult. However, the rest of the results are indicative of a function and an interaction with the proposed target.  Maybe a protein pulldown with the real host could clarify if the interaction is true.

For Figure 1 and the corresponding supplemental: It would be good to indicate the proposed signaP cut site. Maybe the shown sequence is the whole singal peptide? And the sequence should be included in the supplemental for the other 2 effectors.

For Figure 7. I would suggest ANOVA as the statistic test instead of t-test as there are 3 groups which are compared. Please check this also for the supplemental data. T-test can only be used for 2 groups.

For the discussion of the infection data for F. oxysporum and the plants expressing MVLG_05122, might the increased resistance come form the lifecycle differences, biotrophic vs. hemibiotrophic? You might want to discuss this option.

Some typos:

Line 81: proteasome? Is it protease

Line 114: analysis

Line 847: protein complex

Line 852: explanation   

Author Response

Major comments

Dear authors,

The manuscript is well written and has some interesting data concerning new effectors and their host targets form M. lychnidis-dioicae. The research is well done with all the appropriate controls.

The use of the artificial host A. thaliana and the results for the effector MVLG_06175 make the interpretation of the putative host target protein a bit difficult. However, the rest of the results are indicative of a function and an interaction with the proposed target. Maybe a protein pulldown with the real host could clarify if the interaction is true.

We thank the Reviewer for their positive comments and constructive criticisms. While we plan to do pull-down experiments in native host plants, that will be included in a future publication. As stated in the Discussion, Page 17, lines 746-748: “The true identities of A. thaliana plant proteins interacting with the effector proteins in the transgenic plants may require a co-immunoprecipitation assay to reveal.” 

Detail comments

For Figure 1 and the corresponding supplemental: It would be good to indicate the proposed signaP cut site. Maybe the shown sequence is the whole singal peptide? And the sequence should be included in the supplemental for the other 2 effectors.

The signal peptide is now included for each of the effectors, both in Figure 1 and in the Supplemental Figure S2.

For Figure 7. I would suggest ANOVA as the statistic test instead of t-test as there are 3 groups which are compared. Please check this also for the supplemental data. T-test can only be used for 2 groups.

The analysis for Figure 7 has been repeated using a one-way ANOVA. This is shown in the revised Figure 7. As can be seen in the updated figure, the significant difference between host disease susceptibility is found at days 10 and 16 dpi, for plants expressing MVLG_05122DSP-CFP compared to WT or CFP alone; at 22 dpi, this difference is not quite significant (p= 0.055).

For the discussion of the infection data for F. oxysporum and the plants expressing MVLG_05122, might the increased resistance come form the lifecycle differences, biotrophic vs. hemibiotrophic? You might want to discuss this option.

We now discuss this possible difference, Page 19, lines 862-870. Here we emphasize that by 6 dpi, F. oxysporum has likely changed to a necrotrophic phase (Wang et al., 2022) and thus, this could be an additional source of difference from our original expectations regarding MVLG_05122 expression in A. thaliana.

Some typos:

Line 81: proteasome? Is it protease

Corrected

Line 114: analysis

Corrected

Line 847: protein complex

Corrected

Line 852: explanation   

Corrected

Round 2

Reviewer 1 Report

The authors have addressed all the comments in the revised version of the manuscript.

The authors have corrected the manuscript accordingly.